# Co-generation of NaREE(MoO$_4$)$_2$ and REEPO$_4$ in multiple habits by solid-flux crystal growth

Joseph Brian Balta[1,2]*, Megan E. Holycross[2], Buz Barstow[3], Esteban Gazel[2]

1 Lunar and Planetary Institute, Houston, Texas, United States of America, 2 Department of Earth and Atmospheric Sciences, Cornell University, Ithaca New York, United States of America, 3 Department of Biological and Environmental Engineering, Cornell University, Ithaca New York, United States of America

☯ These authors contributed equally to this work.
* jbalta@lpi.usra.edu

## Abstract

Rare Earth Elements (REE) are key to modern technology and industrial processes. They are most used in electronics, although their chemistry enables numerous other applications. Oxides that combine REE, sodium, and molybdenum can be used as catalysts, antibacterial agents, pigments, and even as components in lasers. Although they are heavily in demand, REE supplies are limited in part because the separation of REE from monazite and xenotime (REEPO$_4$), some of the most abundant REE bearing minerals in natural rocks, requires high energy inputs and creates significant environmental hazards. Through an experimental study, here we demonstrate a rapid reaction between REE phosphates and a flux containing molybdenum oxide and sodium carbonate that converts mm-sized REE phosphate into NaREE(MoO$_4$)$_2$ in a period of hours at temperatures <870 °C. This reaction occurs using both lab-grown REEPO$_4$ and naturally occurring monazite as starting materials. The presence of crystalline REEPO$_4$ at higher temperatures (e.g., > 870 °C) limits the grain size of the coexisting oxide. The substantial reactive surface area of the small-grained oxides is advantageous for industrial catalysis processes or for usage as an antibacterial agent. Larger, mm-scale crystalline NaREE(MoO$_4$)$_2$ is produced if mm-scale REEPO$_4$ is not stable in the reaction products at high temperature. Finally, we offer updated details for procedures to grow mm-scale crystals of REEPO$_4$ using the same flux-growth technique, improving the ability to grow those crystals for industrial efforts or for creation of analytical standards.

## Introduction

Rare earth elements (REE) compose the lanthanide group of the periodic table, from lanthanum to lutetium, in addition to Sc and Y, and are subdivided into the Light REE (LREE) and Heavy REE (HREE) groups based on atomic weight [1]. These elements are heavily used in modern electronics, but their supplies are globally limited, with

**Data availability statement:** All relevant data are within the manuscript and its Supporting Information files.

**Funding:** This project was supported by Advanced Research Projects Agency – Energy (ARPA-E) award DE-AR0001341 to B.B, E.G., and M.E.H. This work made use of the Cornell Center for Materials Research (CCMR) Shared Facilities, supported through the NSF MRSEC Program (No. DMR-1719875). The funders had no role in study design, data collection and analysis, decision to publish, or preparation of the manuscript.

**Competing interests:** The authors have declared that no competing interests exist.

over 70% of the REEs produced for industrial use over the last decade originating in China [2]. This supply limitation has prompted the United States Department of Energy to classify REE-bearing compounds as critical minerals and materials (Chu, 2010). Monazite (LREE-phosphate), xenotime (HREE-phosphate), and the LREE fluorocarbonate mineral bastnäsite are the most mined REE mineral resources [3]. The chemical differences between the REE are small, such that any REE-bearing mineral will typically contain some amount of all lanthanides, as well as Pb and the actinides U and Th [1–4].

The development of REE resources begins with ore (a rock or mineral with an economic abundance of one or more elements) mining and extraction, followed by concentration steps that can involve physical sorting, magnetic separation, and flotation aided by the addition of appropriate chemicals [5,6]. Following concentration, the REE minerals are further processed through physical grinding and chemical processing by leaching and roasting, typically using an acid (e.g., sulfuric, hydrochloric, nitric; [e.g., 5, 7]. These reactions are followed by additional precipitation steps that separate various chemical components, including the removal of U, Th, and Pb, all of which have both industrial uses and environmental concerns [e.g., 8–10]. Through these steps the REE are changed from the initial ore material into solid hydroxides, carbonates, chlorides, or nitrates, then finally reduced to a usable metal form.

Processing raw REE resources requires substantial energy and the use of caustic acids, bases, and solvents, all of which have significant environmental impacts and raise the cost of production [2,7–10]. REE refining also results in indirect hazards through creation of large tailings deposits and release of particulate matter into the air [8]. Finally, the solid solution behavior of REE minerals leads to the incorporation of radioactive Th and U in the crystal lattice; these elements and their decay products must be managed to avoid environmental degradation [9,10]. Demand for REE resources is growing rapidly while the financial and environmental costs of accessing REE resources remain high [4], motivating the exploration of less caustic methods and technology for processing REE mineral ores.

Today, the majority of REE resources are extracted from bastnäsite (REE(CO$_3$)F), but this mineral is not the most abundant REE mineral. The REE phosphates monazite (LREEPO$_4$) and xenotime (HREEPO$_4$) are more abundant than bastnäsite, but the use of monazite and xenotime as REE resources has been limited due to the difficulty of extracting REE from phosphate. Industrial processes to break down monazite often involve use of hot acids, which are sufficiently caustic that monazite was described as the "unleachable mineral" [11]. Innovative new phosphate extraction methods are thus required to meet growing demand for REE. Characterizing the efficiency of extraction methods requires careful tracking of REE inputs and outputs in extraction experiments. However, natural crystals of monazite and xenotime commonly have compositions that vary substantially between deposits and include trace elements like U and Th that require special safety precautions [e.g., 12]. Instead, ideal extraction experiments would utilize homogeneous, large-scale, REE phosphate crystals of controlled composition. Here we describe experiments designed to refine existing methodologies for growing REE-phosphate crystals

from flux in high-temperature furnaces. Early experiments yielded NaREE(MoO$_4$)$_2$ instead of REEPO$_4$ and later experiments co-generated both phases. In this paper, we characterize procedures for forming both phases through controlled experiments. While our primary goal was to generate pure crystalline phases as inputs in subsequence REE-extraction exercises, both compounds have important technical uses, and their synthesis is of interest to engineering industries.

Traditional methods for growing monazite involve flux melting with Pb-bearing mixes, but Pb can substitute into the monazite structure and contaminate the mineral chemistry. [13,14] demonstrated techniques for growing monazite from Pb-free mixes using a flux of MoO$_3$ mixed with either Na$_2$CO$_3$ or Li$_2$CO$_3$. They grew crystals of each REE by melting the material at temperatures of 1280°C or 1350°C for each flux type, respectively. However, particularly for the HREE in xenotime, [13] found that the yields of crystals grown were small compared to the initial amount of REE in their experiments, implying the existence of an additional reaction active and a phase generated as an REE host during crystal growth.

If other elements, including transition metals, are present while REE are annealed with heated carbonates, the final reaction product can incorporate both components as we observe in our experiments. For example, studies have produced complex oxides containing alkali elements, molybdenum, and REE through hydrothermal reactions, catalysis with EDTA, or through simple solid-state reactions [e.g.,15–20]. Oxides of this format, in particular NaREE(MoO$_4$)$_2$, have been shown to have numerous industrial applications including: photocatalysis and use in environmental remediation [21–23], luminescence and use in light-emitting diodes [17,19,24,25], pigments [15,21], lasers [26], and use as a photoanode [27]. Related compounds may also have antibacterial properties [22,23,28]. Synthesized alkali-REE-oxides have various crystal and aggregate habits, including octahedra [22], rugby ball shaped [17], microspheres 19], 1-D rods [28], microflowers [21], platy [28], and "irregular" [22,23] depending on the synthesis method employed. The variety of textures and applications of these materials suggests that generation of new morphologies is a valuable target for scientific study.

Despite numerous studies, many properties of alkali-REE-metal-oxides remain unknown. Previous synthesis studies were designed to yield nanometer to micrometer scale crystals; for example, the rugby-ball shaped oxides of [17] were synthesized using EDTA to fix the particle size. Alternatively, the small scale of the oxide crystals synthesized in [15] was hypothesized to be controlled by reactions between the alkali-REE-molybdenum oxides and crystalline phosphates that limited growth of the oxide grains [15]. However, the reactions in [15] have not been fully characterized; in particular, the grain sizes and habits of the oxide grains were not explicitly observed, and the literature does not contain methods for producing mm-scale alkali-REE-metal-oxide crystals through Ostwald ripening [21].

It also may be advantageous to chemically refine REE oxides from ores using molybdenum compounds. Some crucibles used for high-temperature REE redox processing contain molybdenum [5], likely chosen because its low reactivity with liquid REE metals limits contamination [29,30]. Therefore, use of molybdenum at other steps in the REE production process may be economically beneficial, as techniques exist that may be employed to for separate REE from molybdenum and other transition metals [5].

Here we present results of an experimental synthesis study focused on the generation of REE phosphates, with an associated investigation of the alkali-molybdenum-REE-oxide phases formed by reaction between the phosphates and the flux [31]. We give important constraints on the experimental setup required to generate these phosphates, as successful crystal synthesis requires experimental details not previously described in the literature. We further demonstrate experimental techniques that form alkali-molybdenum-REE oxides instead of, or in addition to REEPO$_4$. The size and habit of the alkali-molybdenum-REE oxides generated depends on the progress of a rapid reaction between the REEPO4 starting materials and the molybdenum oxide and alkali carbonate flux. If this reaction goes to completion, it will consume the phosphate and the oxides will coarsen into mm-scale crystals. If the reaction does not go to completion and REE-phosphate is stabilized, the presence of the phosphate limits the size of the oxide, as in [14]. This process creates alkali-molybdenum-REE oxides with crystal habits and grain sizes that have not previously been generated or characterized.

## Materials and methods

This investigation was initiated to synthesize mm-scale REE-phosphate crystals using established Pb-free flux growth methods [e.g., 1,13], but preliminary experiments lacked $REEPO_4$ phases. These initially failed experiments led us to vary experimental and sample preparation procedures, resulting in a detailed characterization of flux-growth techniques for REE phosphates and description of reaction by-products. While testing variations on the procedures of [13] we adjusted numerous parameters of sample preparation and cooling rate, with additional details in Table 1, but for simplicity in this paper we group our high temperature experiments into four general types based on the processes used and the type of material generated in each. As the main goal of this study was originally to synthesize mm-scale $REEPO_4$ crystals, experiment types 1, 2, and 3 are efforts to refine the experimental techniques of [13] for use in our laboratory. Experiment type 4 was designed based on the results of experiments 1–3, and tests for similar reactions occurring in naturally occurring monazite crystals.

1. Synthesis of mm-scale sodium-molybdenum-REE oxides (determined to be $NaREE(MoO_4)_2$) without coexisting $REEPO_4$.

2. Synthesis of mm-scale $REEPO_4$ crystals and coexisting mm-scale $NaREE(MoO_4)_2$ crystals.

3. Synthesis of mm-scale $REEPO_4$ coexisting with separate powders of $NaREE(MoO_4)_2$.

4. Synthesis of $NaREE(MoO_4)_2$ via reaction of natural monazites ($REEPO_4$) with flux.

Experiment types one through three were conducted in a one-atmosphere furnace, while type four experiments were conducted in sealed capsules at high-pressure in a piston-cylinder apparatus.

Type one, two, and three experiments were conducted in covered Pt crucibles in a one-atmosphere Sentrotech horizontal furnace model STT-1600 with tube diameter of 3.5". Loaded Pt crucibles were placed into a ceramic sample holder cut to fit the furnace tube. Following heat treatment, cooled Pt crucibles were placed in beakers filled with DI water and ultrasonicated over 12–18 total hours to liberate synthesized crystals from the surrounding flux. Crucibles were soaked in DI water overnight when ultrasonic cleaning cycles could not be restarted. All components in the DI water were collected and allowed to settle following each ultrasonic step. Visible mm-scale crystals were separated by hand and stored in vials under air. All other materials were allowed to soak in DI water over a period of 4–8 weeks; exact soaking times are unknown as the insoluble material was only recognized after the material was allowed to sit in water for several weeks and soaking times were thus not recorded. Once the material was recognized and recovered, the DI water was replaced weekly to remove dissolved components, but without disturbing any suspended components that had settled out of the fluid. Exact dissolution times varied between experiments, likely as a function of crystal size. All insoluble materials were then gathered, dried, and stored.

## Experiment type 1: mm-scale oxide crystals

The objective of the type one experiments was to synthesize mm-scale REE-phosphate crystals following the technique of [13]. Our starting materials were mixtures of $MoO_3$ and $Na_2CO_3$ or $Li_2CO_3$ (flux) combined with powdered $REEPO_4$. The flux was mixed at a ratio of 75:25 $MoO_3:Na_2CO_3/Li_2CO_3$ and then initially mixed with powdered $REEPO_4$ at a ratio of 100:2 flux:$REEPO_4$ (all ratios are given by weight and reported as P:F Ratio in Table 1). Flux:$REEPO_4$ mixtures were ground in a mortar and pestle for 15 minutes to homogenize, placed in Pt crucibles and covered loosely with a lid. Each experiment was loaded with approximately 50 grams of flux:$REEPO_4$ mixture. Purchased, reagent-grade $CePO_4$ was used as a starting material in the first set of experiments; later analyses confirmed the structure of the purchased starting material was rhabdophane ($CePO_4 \cdot 6H_2O$). Two experiments of this type were attempted using $Li_2CO_3$ starting materials; the remainder used $Na_2CO_3$. As noted below, the $Li_2CO_3$ experiments produced no recoverable material and thus details are not included in Table 1.

**Table 1. Experimental conditions and results.**

| Run # | Type | Starting Phosphate | P:F Ratio[a] | Loading Temp. | Final cooling | Results |
|-------|------|--------------------|--------------|---------------|---------------|---------|
| NX-01 | 1 | Purchased $CePO_4$ | 2:100 | 25°C | Fast (in air) | 1 mm oxide crystals |
| NX-03 | 1 | Purchased $CePO_4$ | 2:100 | 870°C | Fast (in air) | 1 mm oxide crystals |
| NX-04 | 1 | Purchased $CePO_4$ | 2:100 | 870°C | Slow – removed at 600°C | 1 mm oxide crystals |
| NX-07 | 1 | Purchased $CePO_4$ | 2:100 | 25°C | Slow – to 25°C overnight | 1 mm oxide crystals |
| NX-09 | 1 | Purchased $CePO_4$ | 2:100 | 25°C | Slow – to 25°C overnight | 1 mm oxide crystals |
| NX-10 | 2 | Synthesized $NdPO_4$ | 2:100 | 1140°C | Slow – to 25°C overnight | Mixture of oxide and phosphate crystals |
| NX-11 | 2 | $CePO_4$ (Heat treated) | 2:100 | 1250°C | Slow – to 25°C overnight | Mixture of oxide and phosphate crystals. |
| NX-12 | 2 | Synthesized $NdPO_4$ | 2:100 | 1280°C | Slow – to 25°C overnight | Mixture of oxide and phosphate crystals |
| NX-13 | 2 | $CePO_4$ (heat treated) | 2:100 | 1280°C | Slow – to 25°C overnight | Mixture of oxide and phosphate crystals. |
| NX-14 | 3 | Synthesized $NdPO_4$ | 2:100 | 850°C | Fast (in air) | 1 mm phosphate crystals, fine (<100 µm) oxide powder |
| NX-15 | 3 | Synthesized $NdPO_4$ | 4:100 | 850°C | Fast (in air) | Up to 5 mm phosphate crystals, fine (<100 µm) oxide powder |
| NX-16 | 3 | Synthesized $NdPO_4$ | 4:100 | 850°C | Fast (in air) | Up to 5 mm phosphate crystals, fine (<100 µm) oxide powder |
| NX-17 | 3 | Synthesized $YbPO_4$ | 4:100 | 850°C | Fast (in air) | 1 mm phosphate crystals, low yield, fine (<100 µm) oxide powder |
| NX-18 | 3 | Synthesized $YbPO_4$ | 4:100 | 850°C | Fast (in air) | 1 mm phosphate crystals, low yield, fine (<100 µm) oxide powder |
| NX-19 | 3 | Synthesized $YbPO_4$ | 8:100 | 850°C | Fast (in air) | 1 mm phosphate crystals, extremely low yield, fine (<100 µm) oxide powder |
| NX-20 | 3 | Synthesized $GdPO_4$ | 4:100 | 925°C | Fast (in air) | Up to 5 mm phosphate crystals, fine (<100 µm) oxide powder |
| NX-22 | 3 | Synthesized $YbPO_4$ | 2:100 | 925°C | Fast (in air) | 2-3 mm phosphate crystals, good yield, fine (<100 µm) oxide powder |
| NX-23 | 3 | Synthesized $LaPO_4$ | 4:100 | 870°C | Fast (in air) | Up to 5 mm phosphate crystals, fine (<100 µm) oxide powder |
| NX-24 | 3 | Synthesized $NdPO_4$ | 4:100 | 870°C | Fast (in air) | Up to 5 mm phosphate crystals, fine (<100 µm) oxide powder |
| Ni | 4[b] | Natural Monazite | 10:100 | 1000°C | Rapid (Power off/cools in 10s of seconds) | Rare phosphate crystals, radiating oxide crystals surrounding them |
| Ag | 4[b] | Natural Monazite | 10:100 | 950°C | Rapid (Power off/cools in 10s of seconds) | Pitted and irregular phosphate crystals, surrounded by smaller oxide crystals |

[a] P:F ratio shows measured phosphate:flux ratio by weight.

[b] Experiment type 4 done in a piston cylinder.

In type one experiments, the crucible and sample holder were inserted into the furnace at room temperature. Type one experiments used the following temperature cycling routine: ramp from room temperature at 200°C/h to 1280°C ($Na_2CO_3$ flux) or 1350°C ($Li_2CO_3$ flux); dwell at peak temperature for 900 minutes; cool at 3°C/hr to 870°C. Once the furnace reached 870°C, the crucible was either removed from the furnace and quenched in air (cooling to room temperature over ~10 minutes) or cooled over several hours inside the furnace after turning off the heat (Table 1). Although the melting temperature of this flux has not been constrained and was not noted by [13], the melting temperature of $MoO_3$ is 795°C, of $Na_2CO_3$ is 854°C, and of an intermediate compound $Na_2MoO_4$ is 687°C [e.g., 30]; 870°C is thus expected to be

a temperature where the flux remains fully molten. Because experiments would regularly reach 870°C overnight, unless a cooling plan was imposed, the experiment would dwell at 870°C until the crucible could be removed in the morning. Some experiments were timed to allow removal immediately upon reaching 870°C allowing verification that the dwell time at this temperature is a major factor in the experimental results. Two type one experiments were quenched in water immediately after reaching 870°C, but the combination of large sample masses and low viscosities of the liquid flux led to the sample being ejected violently from the crucible if any water entered and loss of nearly all sample. We caution against this quenching technique.

Visible crystals in type one experiments disaggregated upon extraction from the crystallized flux, breaking from mm to sub-mm sizes. The largest crystals were collected, mounted in epoxy, and polished for LA-ICP-MS analysis on a polishing wheel using silicon carbide pads at coarse grain sizes and polished by hand using diamond powders for 1 μm and below. Additional splits of the disaggregated crystals were separated and re-ground to powder by hand using an agate mortar and pestle for XRD analysis. As discussed in section 3.1: experiment type 1 results, none of the type one experiments produced crystalline phosphate.

### Experiment type two: Mixed oxide and phosphate crystals

The objective of type two experiments was to attempt growth of $REEPO_4$ phases while eliminating large crystals of alkali-molybdenum-REE oxides. We changed the following experimental procedures to attempt to meet this objective: 1) $REEPO_4$ starting materials (including lab-generated, rather than purchased, starting materials) were initially heat-treated before subsequent experiments 2) crucibles were placed into the furnace close to the final run temperature and 3) crucibles were cooled slowly to room conditions after ramping to 870°C.

Half of our type two experiments employed new phosphate starting materials synthesized in the lab following the method of [13]. Weighed aliquots of REE nitrates were dissolved in boiling DI water in a Pyrex beaker. Ammonium dihydrogen phosphate was added to this solution to trigger rapid precipitation of a suspended REE phosphate powder. Solutions were dried in an oven at 95°C to evaporate the liquid. Leftover powdered material was transferred by hand to a ceramic crucible. The loaded crucible was covered, placed in a ceramic kiln, heated from room temperature to 500°C in 1 hour, allowed to dwell for 1 hour, heated to 800°C in 1 hour, allowed to dwell for 4 hours, and removed from heat. Full cooling to room temperature in air took approximately 10 minutes. Purchased $CePO_4$ used in the other type two experiments was subjected to a similar heat treatment. All heat-treated phosphates were mixed with a $Na_2CO_3$ and $MoO_3$ flux in a 100:2 ratio, ground, and placed in Pt crucibles.

Loaded Pt crucibles were placed into the horizontal tube furnace pre-heated to the final run temperature of 1280°C (or nearly so). Inserting the experiment in the furnace above the melting point of the flux ensures rapid melting of the flux will occur; however, inserting a large mass of cold material (loaded crucible) into a hot furnace resulted in significant thermal stresses on the ceramic components (crucible holder and horizontal tubes). Sample loading routines were subsequently adjusted for type three experiments. Type two experiments dwelled at peak temperature for 900 minutes before cooling at 3°C/hr to 870°C. All type two experiments were allowed to cool from 870°C to room temperature over several hours in the ceramic tube after shutting off power to the furnace. Crucibles were removed once the furnace had cooled.

All crystals were liberated from the flux using the ultrasonication routine described previously, collected with tweezers, mounted in epoxy, polished, and analyzed for their chemistry. Type two experiments produced some crystals of $REEPO_4$, but they were intergrown with or rimmed by coexisting oxides (Fig 1c), which we interpreted as a reaction during the final cooling step.

### Experiment type three: Successful phosphate crystal synthesis

The objective of type three experiments was to eliminate the intergrowth of phases to produce only phosphates. We changed the following experimental procedures to attempt to meet this objective: 1) loaded crucibles were placed into the horizontal tube furnace at lower temperatures (850°C to 925°C) to limit thermal stresses to the furnace incurred in type

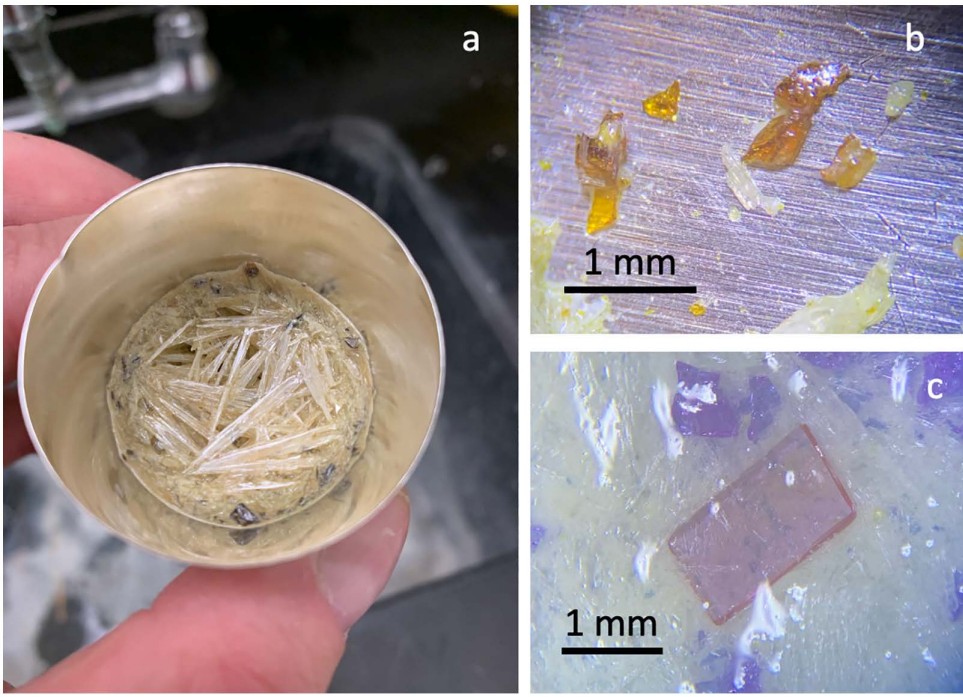

**Fig 1. Images of oxide and phosphate producing experiments.** (a): Platinum crucible after experiment cooling. Cm scale crystals of $MoO_3$/$Na_2CO_3$ flux with dark crystals of Na-Ce-Mo oxide floating on surface. (b) Separated brown crystals of Na-Ce-Mo oxide, variable sizes due to fragmentation, viewed under stereo microscope. Coarse pale crystals are flux prior to dissolution. (c) Euhedral monoclinic grain of Nd-monazite surrounded by fragments of Na-Nd-Mo oxide. Phase ID confirmed by chemical analyses; note color difference between phases. Material photographed just after separation from flux, surrounding material remains in water so that multiple phases could be imaged. Scale bars in (b) and (c) are 1 mm.

two experiments; 2) fast cooling of the experiments during rapid quenching in air; and 3) increased heat treatment time for phosphate starting materials.

The procedure for synthesizing phosphate starting materials used in type three experiments differed slightly from that in the type two experiments. We increased the kiln heating time to 4 hours at 500°C followed by longer dwells at 825°C, with ramp rates as in experiment type two. In sample NX-14 the material was allowed to dwell at 825°C overnight, a full 12 hours compared to the 4 hours in [13]. After seeing improved results in this experiment, we increased the dwell time at 825°C to 68 hours (74-hour full cycle in the kiln) for all subsequent type three experiments; commonly the phosphates were left at temperature over a weekend (Table 1). While this variable was not tested rigorously, we will consider the importance of reaction temperature in the discussion.

Synthesized phosphates and flux were mixed, ground, and loaded into crucibles following previously described procedures. Loaded crucibles were placed into the tube furnace at 850°C or 925°C and heated to 1280°C at 250°C/hour. No difference was observed in the run products for these different loading temperatures. Samples were cooled to 870°C at 3°C/hour, commonly reaching the final temperature overnight, and held at 870°C until the sample could be removed in the morning. Cooling in air took roughly 10 minutes. A fine-grained, insoluble powder was also recovered from all type three experiments after the DI water rinsing step described in Section 2.1: experiment type 1 methods.

Experiment NX-14 utilized a 2:100 ratio of phosphate:flux by weightand this produced crystals that were on the order of 1 mm in size. This ratio was then increased to 4:100 for experiment NX-15, and subsequently this 4:100 ratio was used in the following type three experiments: three $NdPO_4$ synthesis experiments, one $GdPO_4$ synthesis experiment, and one $LaPO_4$ synthesis experiment. However, when the 4:100 ratio was used in $YbPO_4$-bearing experiment, it produced a

limited supply of $YbPO_4$ crystals and a notably larger portion of the fine-grained powder. Later $YbPO_4$ experiments tested flux: REE-phosphate ratios of 8:100 and 2:100, with no other experimental changes implemented.

One additional experiment used $YbPO_4$ starting material with a methodology following the setup of experiment type three, but the experiment was interrupted due to a power loss, leading to a "slow cooling" pattern that more accurately fit the type two experiment setup and thus is classified here. Cooling occurred from approximately 1100°C to 500°C over a period of one hour before the sample was removed and cooling was completed in air. Some mm-scale crystals were recovered following the processing procedure outlined in section 2.1: experiment type 1 methods.

### Experiment type four: natural monazite reaction experiment

The objective of type four experiments was to test whether the reactions characterized in type one and type two experiments would occur with natural monazite starting materials to determine whether this procedure could be used to process monazite ores for extraction. Type four experiments additionally allowed us to constrain the behavior of radioactive Th and U, minor components in natural monazite, during reactions between $REEPO_4$ and flux materials. Type four experiments utilized several pieces of natural monazite purchased from the Platt Pegmatite (also known as the Uranium King mine), Big Creek Pegmatite District, Carbon County, Wyoming US, from a private dealer [32]. This material was verified to be monazite using a WiTec Alpha 300r Raman spectrometer (Section 2.5.1: sample characterization). The surface of the monazite was covered in some areas with crystals of quartz, plagioclase, K-feldspar, and bastnäsite. We selected a portion of a 1 cm x 1 cm x 1 cm monazite grain with the largest surface exposure of monazite and smallest fraction of contaminating accessory minerals for sectioning and use in type four experiments. While this specimen was selected to minimize the abundance of minerals other than monazite, we expect some non-monazite phases were present during the experiments (this is representative of technical applications an industrial setting using naturally occurring materials). The monazite was sectioned with a low-speed diamond saw and shattered into chips mm-scale and smaller with a hammer. A 75:25 mixture of the same $MoO_3$:$Na_2CO_3$ flux used in the type one, two and three experiments was prepared and ground for homogeneity using an agate mortar and pestle.

Flux:REE-phosphate mixtures in all previous experiment types (1–3) were contained in loosely covered platinum crucibles that were open to the furnace atmosphere through the space between the lid and the crucible. Consequently, these mixtures were not fully isolated from their surroundings, leading to the escape of material at high temperatures as demonstrated by colorless crystals of the flux found on the horizontal ceramic tube after experiments. Previous work on natural monazites as well as our own analyses verified the presence of U, Pb, and Th in Platt Pegmatite monazite (Supplementary Table S4 in S1 File). To avoid the possibility of contaminating the lab with these elements, type four experiments were conducted in sealed capsules in piston-cylinder presses in the Experimental Geochemistry lab at Cornell University. The advantage of the piston-cylinder is that the pressure applied to the capsule during the experiments ensures U, Pb and Th do not escape during the experiment. Two capsule compositions were used to test for interaction of the monazite samples with capsule metals. Rods of nickel and silver metals were machined to create capsule "cups" and cut to size. Approximately 50 milligrams of flux material were packed into the drilled capsule, leaving some void space on top. Five milligrams of the crushed monazite were added to the top of the flux using tweezers; it was expected that melting of the flux would allow these crystals to sink during the experiment and this would allow mixing between the crystals and flux. The starting materials were packed into the capsule using a drill blank and covered loosely with a 1 mm metal disk cut from the same metal rod used for the capsule cup.

Filled capsules were loaded into a ½" piston-cylinder assembly. Assemblies consist of cylindrical $BaCO_3$ cell, a cylindrical graphite heating element, crushable MgO spacers, and a ceramic $Al_2O_3$ sheath surrounding the capsule. Temperature was monitored during the experiments with a W/Re type C thermocouple in a ceramic sheath. More details about the piston-cylinder assemblies used in the Experimental Geochemistry Lab at Cornell can be found in [33,34]. Assemblies were loaded into a Rockland Research Corporation piston-cylinder and compressed to reach an estimated pressure of

0.5 GPa; the materials were held cold under these conditions to create a soft seal on the metal capsule overnight. Both samples were heated above the flux melting temperature; Ni capsule experiments were heated to 1000°C while the Ag capsule experiment was heated to 950°C due to the lower melting point of Ag metal. Samples were allowed to dwell for 6 hours and then cooled at a rate of 100°C/hour until they reached 400°C. Samples were quenched by turning off the power, resulting in cooling to 100°C within ~20 seconds. Capsules were removed from the used assembly, mounted in epoxy, cut using a diamond wafering saw, and polished in preparation for analysis.

## Sample characterization

**Raman spectroscopy and X-Ray diffraction.** All samples were initially characterized by optical microscopy. Additional imaging and mineral identification analyses were conducted using a WiTec Alpha300 R confocal Raman imaging microscope at PI Gazel lab at Cornell University. Images were taken using the standard binocular microscope setting and saved using the WiTec Control Five Software. Mineral identification was conducted using a 532 nm (green) laser and processed using the Control Five and Project Five software. Additional mineral identification was conducted using the Bruker D8 Advance ECO powder diffractometer in the Cornell Center for Materials Research. For verification of the identity of oxide crystals, coarse crystals were separated from the flux by hand and re-ground using an agate mortar and pestle. Peak fitting and mineral identification were conducted using the associated Jade analysis software and ICCD database.

**SEM and EMPA imaging and analysis.** Images of both microcrystalline and coarse materials were collected using a Zeiss Gemini 500 Scanning Electron Microscope (SEM) with an Oxford Instruments Ultim Max EDS detector in the Cornell Center for Materials Research (CCMR). Microcrystalline samples were first attached to carbon tape and gold-coated, while mm-scale crystals were only mounted on carbon tape and cleaned using isopropanol prior to imaging. Images of microcrystalline and coarsely crystalline materials were collected with an accelerating voltage of 15 KeV; voltage was increased to 30 KeV for EDS analyses of heavy elements. For experimental charges, samples were cleaned with isopropanol and carbon coated prior to analysis. EDS maps were collected covering full exposed charges to locate rare phases, and spot analyses were conducted to verify mineral chemistry. EDS maps were again conducted with accelerating voltages of 30 KeV.

Electron Microprobe (EPMA) WDS analyses were conducted on experimental charges using the Cameca SX5-Tactis probe at the American Museum of Natural History in New York City. Polished experimental charges were cleaned with ethanol and carbon coated prior to analysis. Analyses were conducted using an accelerating voltage of 15 KeV and a current of 40 nA. Standards used include Na on albite, Ni on Ni metal, Y and P on $YbPO_4$, Mo on Mo metal, Ag on Ag metal, La on $LaPO_4$, Ce on $CePO_4$, Pr on $PrPO_4$, Nd on $PO_4$, Sm on $SmPO_4$, Eu on $EuPO_4$, Gd on $GdPO_4$, Pb on Pb metal, and Th on Th metal. Reported detection limits varied between 110 ppm for P to 1360 ppm for Pb, with most REE between 700–800 ppm. Analytical standard deviations vary from 0.05 wt. % on Pb to 0.2 wt. % on Ce, except for Mo which reports an average standard deviation of 0.5 wt. %. Microprobe spots were chosen based on previously collected SEM maps to verify the chemistry of experimentally produced phases. Totals typically did not approach 100%, reflecting the abundance of low mass, high mass, multiple-component, and volatile-bearing phases in the charges that were not included as part of our EPMA analytical routine. However, the analyses were sufficient to confirm the identity of various phases present in completed experiments and to allow order-of-magnitude estimates of element partitioning.

## LA-ICP-MS

Select samples of coarsely crystalline phosphates and oxides were mounted in epoxy, polished, and analyzed using an Agilent 8900 ICP-MS/MS and ESI NWR 193HE laser in the Cornell Mass Spectrometry Facility (CMaS). Analyses were conducted under the following laser conditions: RF power 1250–1300 W, Neb gas flow 75 L/min, sample depths 4–6 mm, rep rates of 10 hz, fluences of 5 J/cm$^2$, 30s ablation times, and spot sizes of 100–50 μm. The larger spot sizes were found

to cause some elements to hit the detector maximum requiring smaller spots for those materials. NIST 610 and NIST 612 glasses were used as standards and Durango Apatite was measured as an unknown to verify data quality, and $^{31}P$ was used as an internal standard. Data reduction was done in Iolite software package.

## Results

### Results of experiment type one

Type one experiments are defined by 1) use of purchased $CePO_4$ starting materials and 2) a lack of heat treatment for the phosphates. No consistent pattern of loading and unloading the crucibles at temperature was used; both slow cooling and slow heating steps were present, and only experiment NX-03 was loaded into a preheated furnace and cooled rapidly in air as happened in later type three experiments. Type one experiments using the $Na_2CO_3$ flux produced brown crystals floating at the top of the crystallized flux (Fig 1a). The synthesized $CePO_4$ of [13] was described as green, indicating that the brown crystals in our type one experiments were likely not $CePO_4$. Brown crystals were initially 1–5 mm in size but disaggregated when they were handled with tweezers, typically breaking into fragments that were 1 mm in size and smaller (Fig 1b). Optical examination of the brown crystals embedded in the flux showed that there were numerous internal fractures in the crystals that promoted disaggregation. XRD, LA-ICP-MS and SEM analyses confirmed the brown crystals to be $NaREE(MoO_4)_2$ (Fig 2). No phosphate minerals were detected in the flux or the crystalline materials at the detection limit of the XRD or during SEM or LA-ICP-MS analysis. The end state of the phosphate is unknown but could have been removed during the disposal of soluble flux components. The lack of $REEPO_4$ in this experiment design motivated the redesigns for experiment types two and three.

Two type one experiments were performed using a flux containing $Li_2CO_3$ instead of $Na_2CO_3$. No coarse-grained materials were observed in $Li_2CO_3$-bearing experiments, and no material was recovered after the DI water ultrasonication. If an insoluble Ce-residue was produced in these experiments, it was fine-grained enough to remain in suspension and lost during ultrasonication. Further tests using $Li_2CO_3$ fluxes were not performed after these results.

### Results of experiment type two

Type two experiments are defined by 1) a near-instantaneous heating process (samples were inserted in the furnace heated to $T \geq 1140°C$ 2) heat treatment of phosphate starting materials and 3) slow cooling after reaching 870°C. All type two experiments produced coexisting mm-scale $REEPO_4$ and mm-scale alkali-REE-Mo oxide crystals. Material resembling the fragile mm-scale crystals of experiment type 1 was observed in experiments NX-11 and NX-13, which used purchased $CePO_4$ starting materials, along with mm-scale crystals that had similar colors but did not fracture. Experiments NX-10 and NX-12 used laboratory synthesized $NdPO_4$, and run products include rare purple crystals with a monoclinic habit ($NdPO_4$) along with more numerous fragments of differently colored purple crystals that fractured consistent with grains observed in experiment type one ($NaNd(MoO_4)_2$; Fig 1c). The slow cooling step implemented in type two experiments produced elongate and interlocking flux crystals surrounding REE minerals. The largest REE crystals in experiment NX-11 were polished and found to be intergrowths of $CePO_4$ and $NaCe(MoO_4)_2$; in experiment NX-13 analyses showed that the grains had cores of $CePO_4$ and rims of $NaCe(MoO_4)_2$. Similar rims were observed in sample NX-21, which cooled rapidly from high temperature due to power loss.

### Results of experiment type three

Type three experiments are defined by: 1) an intermediate-rate heating process (samples were inserted in the furnace heated to 850−925°C); 2) increased heat treatment time for phosphate starting materials; and 3) fast cooling after dwell at 870°C. All type three experiments produced mm-scale $REEPO_4$ crystals accompanied by a fine oxide powder (Fig 3). The habit and color of the $REEPO_4$ in type three experiments matched those described by [13]. The first experiment in our

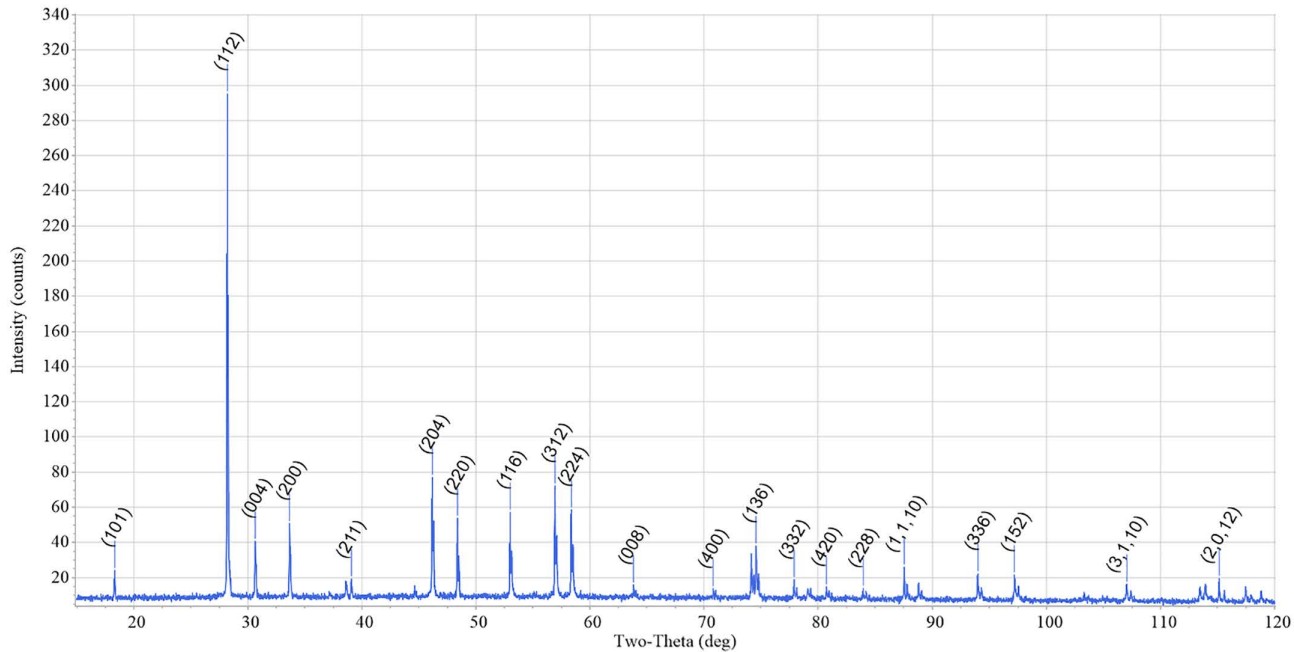

**Fig 2. X-Ray Diffractogram/spectrum analyses of generated oxide crystals.** Peaks identified as $NaCe(MoO_4)_2$ by internal XRD Jade analysis software using ICCD database.

type three series, NX-14, precipitated well-formed homogeneous $NdPO_4$ crystals. However, the crystal phosphate yield was low and grain sizes were small (maximum grain size ~1 mm). We attempted to maximize phosphate crystal yields in a series of subsequent experiments (NX-15 to NX-20) by increasing the dwell time for the Nd-phosphate pre-heating step and by mixing new starting materials with a higher ratio of phosphate:flux. These adjustments yielded a larger proportion of massive $NdPO_4$ with the largest crystal up to 5 mm in diameter (Fig 3). For $NdPO_4$, 3 matching experiments were performed with this setup (NX-15,16,24) and each yielded identical results, with similar crystal sizes and yields of large crystals. The same procedure, when applied to $LaPO_4$ and $GdPO_4$ led to similar crystal sizes and yields of large crystals, allowing us to state that this technique was successfully repeated 5 times across the light and middle rare earths, with similar phosphate crystal yields. As in [13], the main difference in the results is crystal morphology, as the crystal morphology varied mostly due to the choice of REE in the synthesis experiments.

While La-, Nd-, and $GdPO_4$ runs produced consistent results, $YbPO_4$ experiments had noticeably worse phosphate crystal yields when the 100:4 flux:$YbPO_4$ starting material was used. Changing the flux: phosphate ratio to 100:8 decreased the phosphate crystal yield further, producing a limited number of crystals ≤1 mm diameter. These crystals required sieving to separate them from the surrounding insoluble powder, the abundance of which was noticeably higher than in the experiments with the LREE. Reducing the flux:$YdPO_4$ ratio to 100:2 led to increased crystal production and larger crystal sizes (Fig 3).

All type three experiments produced fine-grained insoluble residues that were isolated from the flux. EDS analyses confirmed that the chemistry of the fine-grained residues is consistent with crystalline $NaREE(MoO_4)_2$. The habits of these crystals are commonly elongate with skeletal growth patterns and diameters <1 μm and up to 100 μm (Fig 4). Oxides with twinned and needle-like habits were also observed, including needles with gentle bending to them. Analyses of the $NaYb(MoO_4)_2$ powder from NX-22 revealed smaller needles appear to grow off a single source needle. We note that the grain size of $NaYb(MoO_4)_2$ was smaller, and the needle-like habit was more abundant compared to REE-Mo oxides in our

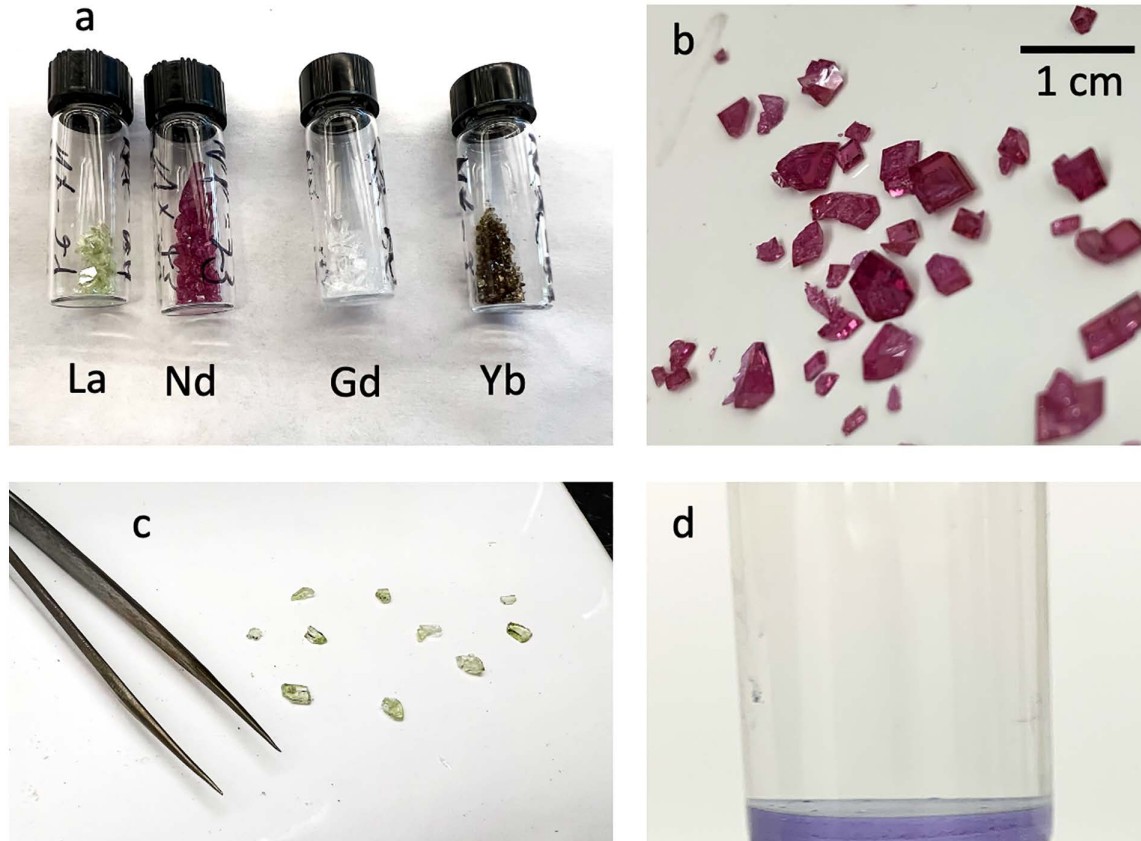

**Fig 3. Materials produced in type 3 synthesis experiments.** (a) shows 4 colors of phosphate crystals tested here, with full recovered crystal batches from each, labeled by REE variety. (b) full crystal recovery from Nd-phosphate run NX-15. (c) full crystal recovery from La phosphate run NX-23. Tweezers for scale in (c). (d) Coexisting powder from Nd phosphate run NX-15 after being allowed to fully settle out; vial is filled with water and layering was produced during settling.

other type three experiments (Fig 4 c-d). Fine-grained alkali-REE-oxides were not observed by [13] but their presence in our experiments is consistent with the results of [15], who noted their synthesized phosphate crystals coexisted with a fine powder of $NaREE(MoO_4)_2$.

## Results of experiment type 4

Type four experiments are defined by 1) use of natural monazite starting material and 2) heat treatment at high-pressures in a piston-cylinder apparatus. Type four experiments are labeled by capsule type in Table 1 (Ni or Ag). Capsules in both experiments were sealed throughout the duration of the experiment. The loaded flux remained in the capsules after completed experiments were sectioned and impregnated with epoxy. Both experiments were likely saturated with $CO_2$ vapor at high pressure-temperature conditions, evidenced by the presence of mm-scale void spaces in the flux. Some areas of the flux could be polished for chemical analysis, particularly in the Ag capsule experiment where the flux is well exposed away from the monazite crystals (Fig 5).

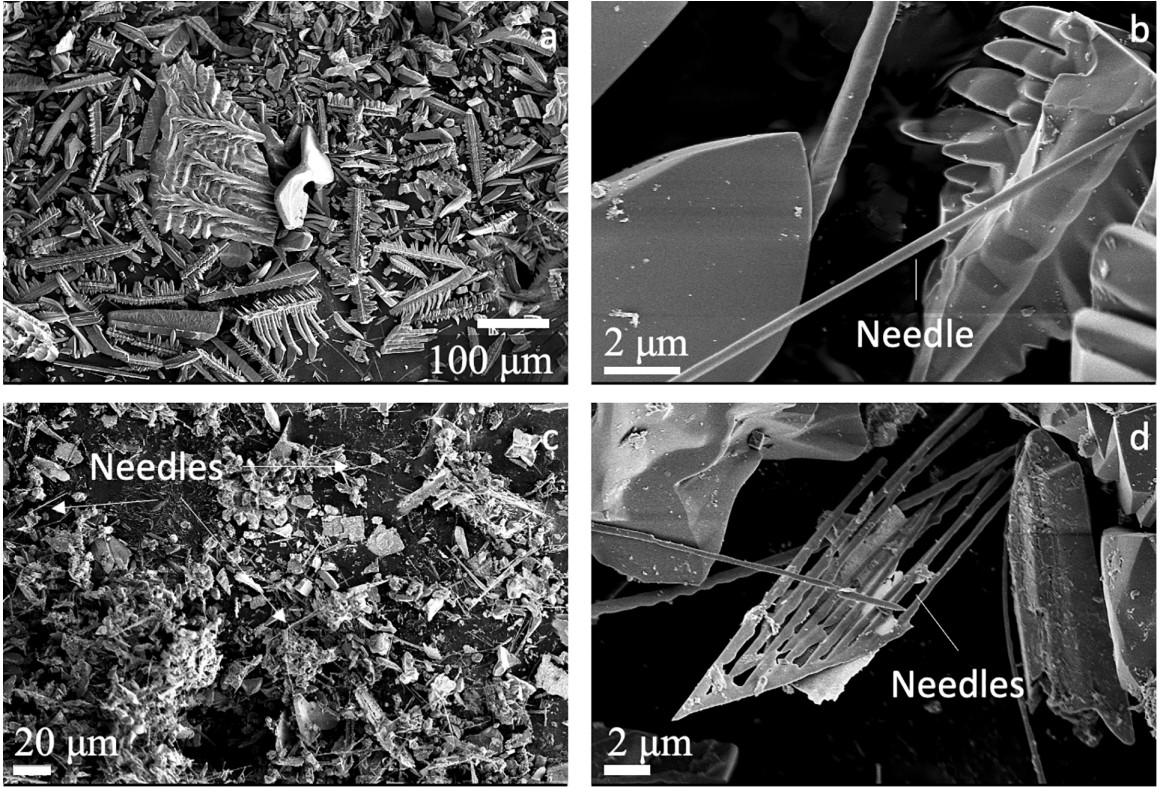

**Fig 4. SEM InLens imagery of fine-grained oxides that coexist with REE phosphates.** (a) and (b) Nd oxides from experiment NX-15. (c) and (d) Yb oxides from experiment NX-22.

Type four experiments preserve the in-progress conversion of natural monazite to $NaREE(MoO_4)_2$ in the flux medium. In the Ni capsule experiment, rounded to irregular monazite crystals approximately 100 µm in size are surrounded by lathes of $NaREE(MoO_4)_2$ radiating away from the central phosphate crystal. $NaREE(MoO_4)_2$ lathes are up to 500 µm in length. Fine µm-scale clusters of flux crystals grew alongside conglomerates of larger phosphate and oxide crystals. We identified two clusters of crystals consisting of oxide lathes emanating from phosphates; one at the bottom of the capsule and the other the middle of the capsule (Fig 5a), fully surrounded by the flux and not visibly in contact with the capsule edges or base (although this could not be confirmed in the third dimension). Dark/low contrast material surrounding these crystals is poorly polished flux and epoxy holding materials together. Although microprobe totals were poor (typically >100 wt. %), we were able to verify that all $NaREE(MoO_4)_2$ oxides in the Ni-capsule experiment contained approximately 10 wt. % NiO and had lower abundances of MoO compared to those in experiment types one through three (Supplementary Tables S1-S4 in S1 File). This suggests the Ni from the capsule may substitute for Mo, yielding $NaREE((Mo,Ni)O_4)_2$, enabled in this experiment by the matching 2+ charge of oxidized Ni. The flux matrix also contained grains of pure Ni. No obvious fractionations of any REE were detectable at the limits of the microprobe analyses, i.e., the initial element ratios and abundances of the monazite crystals and the oxides were preserved in the oxides with no measurable deviations given the analytical errors.

Large (500 µm) crystals of monazite with irregular edges and pitted surfaces were found in the Ag capsule experiment. Monazite crystals are surrounded by clouds of smaller crystals of $NaREE(MoO_4)_2$, with sizes ranging from 1–50 µm (Fig 5b). Contamination by capsule materials was limited in this experiment compared to the Ni capsule experiment, with

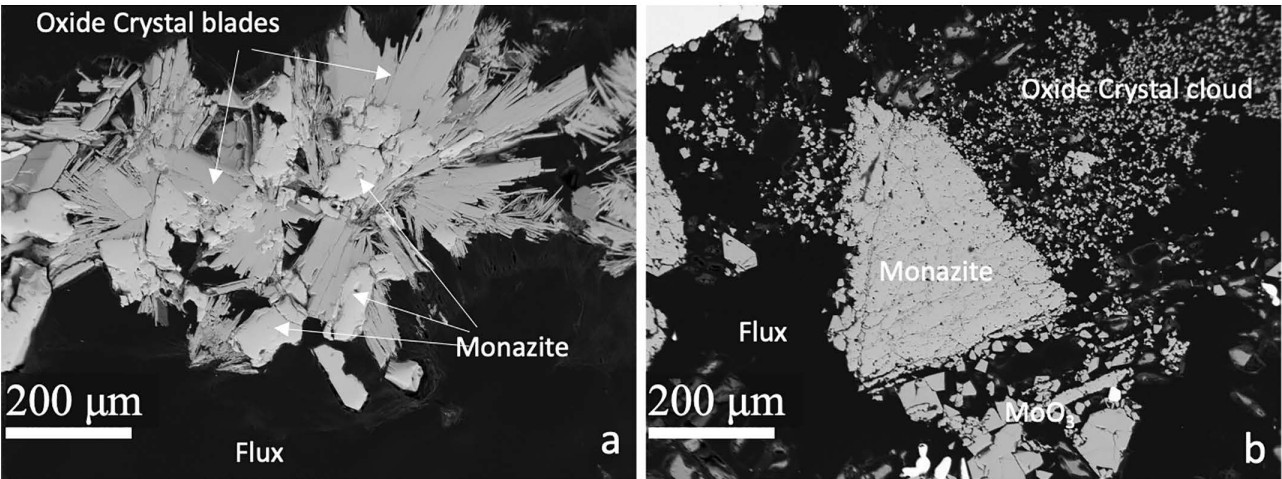

**Fig 5. Electron Microprobe Backscattered Electron Images of natural monazite reaction experiments.** (a) Experiment performed in Ni capsule (not pictured). Rounded/irregular monazite grains are surrounded by lathe-like growths of Na-REE-Mo oxides, with measurable Ni substituting for Mo in all measured grains. Dark areas surrounding grains were filled with $MoO_3$-$Na_2CO_3$ flux and are difficult to polish, likely due to carbonate fluids in pore space. (b) Experiment done in Ag capsule (shown at top left). Large monazite grain at center of frame shows rounded edges and unstable, irregular texture, with smaller grains of Na-REE-Mo oxides forming surrounding it. Some larger lathes of $MoO_3$ grown from the flux are shown at the lower right.

the highest Ag abundances in the newly grown oxide crystals measured at 0.4 wt.%. Compared to the Ni capsule, larger lathes of the flux remained intact after polishing and were measured as nearly pure $MoO_3$. Like the Ni capsule experiment, no obvious fractionations between the measured REE abundances and ratios in the phosphate and the resulting oxide crystals were observed.

## Discussion

### Refinement of phosphate growth procedures

This work began as an effort to grow large phosphate crystals following previously published procedures for crystal growth experiments using a Pb-free flux. Our type one experiments failed to produce REE-phosphates, but adjustments to our experimental methodologies yielded large REE-phosphate crystals in subsequent runs. Here we discuss the experimental issues we encountered and our recommendations for procedural improvements.

Although we purchased experimental starting materials labeled $CePO_4$, the reagent we received was not pure monazite but rhabdophane instead. This was consistent for $CePO_4$ purchased from two different suppliers, and we noted that the grain sizes in each batch were different. The reaction that produced alkali-REE-Mo-oxide appears to run to completion if the experimental starting material is rhabdophane rather than pure $REEPO_4$ monazite. This is evidenced by the results of experiment NX-03, where the rhabdophane-flux starting materials were loaded into a hot furnace and cooled rapidly in air. Our type three experiments suggest these run conditions should yield large crystals of $REEPO_4$, but NX-03 produced only the oxide phase. Thus, we suggest caution about the use of purchased reagents in flux-growth experiments without verifying their compositions. Further, we suggest all starting materials for crystal growth experiments (purchased or otherwise) be heat treated, regardless of previous chemical identification.

In most of the type 3 experiments, phosphate starting materials were heat treated for longer durations than recommended by [13]. This longer heat treatment step produced our best results. We recommend a heat treatment of >24 hours at 825°C for phosphate starting materials and note that a 72-hour heat treatment was used in experiments that produced the greatest yield of mm-scale phosphate crystals. As discussed below in section 4.3: Nature of the phosphate-oxide

reaction, heat treatment of starting materials at >825°C may lead to improved phosphate crystal yields, particularly in the HREE.

Reactions between the phosphate and the flux occurred in all experiments that utilized starting materials that were not pure REEPO$_4$ (type one) and in experiments where the flux+phosphate mixture was exposed to T < 850°C for extended time periods (type one and type two). These reactions produced large crystals of NaREE(MoO$_4$)$_2$ and consumed the phosphates. We infer these reactions primarily occur at lower temperature (<850°C), since large oxide crystals were not found in type three experiments, which were inserted into a pre-heated furnace at T > 850°C and quenched rapidly. Our type one experiments typically employed a heating rate of ~200°C/hour, allowing several hours for the reaction between liquid flux and phosphate during heating of the experiments loaded into a cold furnace. Type two experiments were inserted into a pre-heated furnace but experienced slow cooling from the 870°C stopping temperature to room conditions. This cooling rate was not precisely quantified but likely resulted in crystallization of the liquid flux in under 2 hours once power was cut to the furnace. This time was apparently sufficient to allow much of the monazite crystallized at high temperatures to convert into large oxide grains during cooling. In type 3 experiments, the sample was allowed to dwell at temperatures of 870°C until the experiment could be removed from the furnace, and no conversion to oxide was observed on the crystal edges, indicating that holding at this temperature is the key step to prevent reaction to oxide. We recommend experiments be heated and cooled rapidly to preserve phosphates and limit reaction with the flux at temperatures below ~870°C. We identified an insoluble powdered REE-oxide residue in type three experiments that produced mm scale REEPO$_4$ crystals (Figs. 3-4). REE-oxide residues were not identified in the experiments of [13] but their presence in our experiments is consistent with the low phosphate crystal yields they reported in the HREE. We recommend separating this material by slowly dissolving away all solid flux crystals to fully characterize the REE material or to conduct mass balance calculations. [15] generated similar material using a solid-state reaction and suggested it could be used as a potential pigment. However, we recommend caution when handling this material as SEM investigation revealed the presence of a needle-like (like asbestos) habit that requires further investigation [e.g., 35].

## Generation of NaREE(MoO$_4$)$_2$ oxide habits and potential uses

NaREE(MoO$_4$)$_2$ phases have been shown to have numerous potential industrial applications; for example, [21] demonstrated these phases are useful in environmental remediation. We generated and collected insoluble powdered residues of NaREE(MoO$_4$)$_2$ in all synthesis experiments where mm-scale phosphate crystals were present and stable at the end of the experiment (experiment type 3). There are potential benefits in generating NaREE(MoO$_4$)$_2$ using flux-reactions with naturally occurring phosphates, including lower costs and increased production yields.

NaREE(MoO$_4$)$_2$ has previously been suggested as a potential pigment [15]. SEM imaging of these powders demonstrated skeletal crystal habits and grain sizes ranging from 1–100 μm were common (Fig 4). The grain habits and sizes for La-, Nd-, and Gd-Na-Mo oxides were similar, while the grain size was smaller for NaYb(MoO$_4$)$_2$. The habits of all these materials included large surface areas as seen in the SEM images in Fig 4. Thus, for uses where reactive surface area is critical, such as antibacterial or reaction catalysis applications [21,23,28]; the materials produced in these phosphate growth experiments may be appropriate and straightforward to synthesize.

The slow heating and slow cooling experiments (type one and two) demonstrate that the modified molybdenum oxide and sodium carbonate flux crystal growth methods of [13] employed for REE phosphates may also successfully allow growth of mm-scale crystals of NaREE(MoO$_4$)$_2$. We note, however, that use of Li$_2$CO$_3$ as a flux does not enable growth of mm-scale alkali-Mo-REE oxides, in contrast to experiments that utilized Na$_2$CO$_3$. Oxides containing Li instead of Na were specifically noted by [26] as potentially useful in lasers, and thus alternative methods of growing those crystals may be necessary.

For natural monazite samples, our piston-cylinder experiments (type four) demonstrate the feasibility of producing NaREE(MoO$_4$)$_2$ in bulk quantities. The habits of these crystals were no longer skeletal, but instead elongate and tabular

in the Ni capsule and equant in the Ag capsule. However, we also note that elements such as U and Th, which are commonly present in naturally occurring monazites, are transferred to the newly grown oxide crystals and thus would need to be accounted for in any process using these materials.

**Nature of the phosphate-oxide reaction and potential for large scale industrial uses**

Using carbonates to "crack" REE-phosphate or REE-fluoride minerals and convert the REE into oxide form is a method of processing some REE ores [2,36,37], commonly referred to as "roasting" with carbonates. The presence of carbonates in the flux in this experimental method suggests commonality between the reactions in [13] and industrial carbonate processing. In carbonate roasting, the anions (either fluorine or phosphate) bonded to the REE exchange with sodium carbonate at temperatures from 500–700°C. This exchange yields sodium phosphate or sodium fluoride, REE oxide, and carbon dioxide gas as products. The REE oxide is then dissolved in an acid and reduced through redox or electrical process [2 and references therein], commonly with additional steps required to separate and purify individual REEs [2,37]. Carbonate roasting techniques are not often applied to compounds where standard processing methods such as froth flotation can concentrate the ore to a sufficient abundance (~30%) [5,7] but have been useful for separation in materials where REE concentrations are lower or the REE bonding environment is unusual, like recycled materials or previously-processed mine tailings [37].

Our experiments demonstrate a reaction between REE phosphates (e.g., monazite and xenotime) and a flux containing $Na_2CO_3$ and $MoO_3$ that produces crystalline $NaREE(MoO_4)_2$ as a reaction product. This reaction occurred during our experiment heating and cooling steps, and appears to reach completion in roughly one hour at temperatures below 870°C. A schematic reaction is shown here:

$$REEPO_4 \ + \ 2MoO_3 \ + \ 2Na_2CO_3 \ = \ NaREE(MoO_4)_2 \ + \ 2CO_2 \ + \ Na_3PO_4{}^{inferred} \tag{1}$$

As noted in the results, when this reaction ran to completion no phosphorus was observed after the experiment, so the inferred $Na_3PO_4$ represents a plausible soluble component that could solve this mass balance issue but would not be detectable after dissolution and removal of the remaining flux. Based on the preservation of monazite crystals in the type 3 experiments and the growth of mm-scale crystals during the cooling process, we believe this reaction is not active at high temperature when monazite or xenotime is the stable phase. [38] noted that the hydrous, hexagonal REE phosphate rhabdophane forms when precipitation from solution is done below 90°C, as in all our synthesis reactions. They then showed that rhabdophane transitions to an anhydrous hexagonal phase between 100 and 400°C intermediate between rhabdophane and monazite, and then finally to true monazite at temperatures as low as 400°C for $LaPO_4$, but as high as 900°C for $DyPO_4$. The transition to the xenotime structure for $YbPO_4$ was similarly noted to occur at temperatures above 860°C, presumably from a similar hexagonal phase. Thus, the heat treatment steps tested here may be insufficient for full conversion of the heaviest REE to the high-temperature monazite or xenotime structure. This may explain the low phosphate crystal yields in the HREE experiments here and in the work of [13]. However, we were still able to produce experiments with $YbPO_4$ crystals at the conclusion of experiments, so some conversion to the xenotime structure must have occurred during our heat treatment step at 825°C or during heating of the full experiments. If the starting phosphate was a mixture between xenotime and the hexagonal phase, as may have occurred with $YbPO_4$, there would be greater conversion to $NaYb(MoO_4)_2$ during heating, leading to low HREE phosphate yields in our experiments. Future work applying this technique to growing $HREEPO_4$ crystals should include tests of whether higher dwell temperatures at this step lead to improved phosphate crystal yields.

The intermediate hexagonal structure is a candidate for the $REEPO_4$ phase that reacts with the flux to form the $NaREE(MoO_4)_2$ powder found in all type three experiments. This reaction must occur rapidly enough that some oxide is generated during the initial heating as the hexagonal phase would not be stable once the crucible equilibrated with the

furnace temperature and phosphate crystals were stable and growing during the run. Thus, the brief crucible heating is the only potential time when the generation of this oxide could have occurred in our type one experiments.

[39] mixed $LaPO_4 \cdot H_2O$, presumably in the rhabophane structure, with $Na_2CO_3$ and explored the reactions as temperature was increased. Unlike these experiments, they report no heat treating to dehydrate their material and form the monazite structure prior to carbonate roasting. Their experiments showed a dramatic decrease in reaction temperature associated with ball milling, 30 minutes of which dropped the particle size to an average of a few micrometers and led to almost complete reaction of the $LaPO4 \cdot H_2O$ by a temperature of 700°C. While grain size was not monitored in these experiments, and the structure of their phosphate is likely to have been more readily reacted than ours, their experiments do support our suggestion that rapid reaction between a phosphate and carbonate flux is possible during the time required to heat our samples. Ball milling was not done on our powders; all samples were ground in a mortar and pestle to ensure homogeneity after the carbonate and $MoO_3$ were mixed with the phosphate, but we expect that this was much less intensive than 30 minutes of ball milling as used in [39]. The grain size produced through chemical precipitation was not tested, and it is unknown whether the grain size could vary between different REE, but this did not appear to be a major variable in any of our experiments.

The nature of the reaction that occurred during cooling in the type two experiments is more challenging to assess as the monazite structure is not expected to revert to the hexagonal form during cooling. However, as demonstrated in the slow-cooling experiments (type 2) and the elevated pressure experiments (type 4), this reaction is rapid enough to convert mm-scale crystals of monazite to oxide within a few hours if the material is cooled to temperatures in the range of 400°C, the lowest temperature reached in the type 4 experiments. Therefore, it is possible that either the monazite reacts with the flux directly at these temperatures or alternatively there could be a structural conversion at the crystal surface that produces a reactive phase and consumes the grains.

Regardless of the reaction pathways, our experiments demonstrate a process with industrial potential. A reaction process that converts monazite to an oxide that is easier to process than the phosphate, which occurs quickly, runs to completion, which produces a stable material that can be separated using simple solution chemistry, and which uses materials already tested for REE processing offers potential industrial benefits for monazite processing. While this conversion does not deal with contaminants such as U or Th, and additional steps may be required to remove the transition metal from the final oxide, previous literature suggests that the inability to break down monazite has been a major impediment to its use as an REE source. It is possible that this method, roasting in the presence of a transition metal such as molybdenum, could be a method of processing either recycled REE or minerals that are typically difficult to access, as shown here with monazite. Our experiments show that generation of $NaREE(MoO_4)_2$ represents such a potential intermediate step for monazite processing.

## Conclusions

In this paper we have explored many details of the process of growing mm-scale crystals of $REEPO_4$ monazite and xenotime using $Na_2CO_3$-$MoO_3$ fluxes. While a previous procedure existed for growing crystals to use as starting materials in experiments or as standards for analyses, our efforts revealed important details needed included in the previous procedures to make them reliable in our lab and in future attempts at monazite synthesis. These details include requiring rapid heating and cooling of the sample from elevated temperature to avoid reactions between the phosphate and the flux material, which we also argue represents a more general relationship between phosphate minerals and carbonate-molybdenum fluxes. We demonstrated that this reaction occurs rapidly and produces a fine-grained powder of $NaREE(MoO_4)_2$ in all experiments, which leads to yields of $REEPO_4$ crystals that always represent <100% of the initial REE. We also showed this material forms skeletal and a needle-like habits; while numerous morphologies have been demonstrated in the literature for these materials, the other morphologies were commonly produced in small quantities and from synthetic materials. In contrast, our technique produced substantial quantities of $NaREE(MoO_4)_2$ and could

 

be applied for conversion of natural materials into this oxide. $NaREE(MoO_4)_2$ has numerous industrial applications, and our experiments produced substantial amounts of material with large reactive surface areas; while many of the previous methods produced reactive surface areas, this technique offers the advantage of working with available natural starting materials and produces material without sacrificing the reactive surfaces that could be used for catalysis. Consequently, the methods detailed here could be used to produce $NaREE(MoO_4)_2$ quickly and in sufficient quantity for industrial use. $REEPO_4$ starting materials may be completely converted to alkali-Mo oxides and grow to mm-scale crystals using our new technique. The formation of $NaREE(MoO_4)_2$ phases may also be explored as a potential intermediate step in new methodology for processing natural monazite into a usable REE resource, and future work should explore reactions to produce this compound from natural minerals.

## Supporting information

**S1 File.** TableS1.xslx: EDS analyses of oxide crystals formed in initial experiments, given as raw EDS analyses. TableS2.xslx: LA-ICP-MS analyses of experiments NX-11 and NX-12 showing intergrown oxide and phosphate phases. Standard analyses included, processed and output using Iolite software package. TableS3.xslx: LA-ICP-MS analyses of phosphate crystals from experiments NX-14 through and NX-23 showing phosphate phases, with all other REE analyzed to test for cross contamination between experiments and to measure degree of Molybdenum contamination. Phosphorus commonly off scale due to it being a major element in all phases. Standard analyses included and labeled, processed and output using Iolite software package. TableS4.xslx: LA-ICP-MS analyses of Wyoming monazite starting materials. Standard analyses included, processed and output using Iolite software package. TableS5.xslx: EPMA analyses of experiments Ni and Ag, conducted in piston cylinder with Ni and Ag capsules respectively. All rows labeled, standard analyses included, full microprobe output including standard ID and probe settings listed.
(ZIP)

## Acknowledgments

The authors appreciate the assistance of Lyndsey Fisher of Cornell University with LA-ICP-MS analyses, Keiji Hammond and Nicholas Tailby of the American Museum of Natural History with electron microprobe analyses, Malcolm Thomas of Cornell University with SEM analyses, and Mark Pfeiffer of Cornell University with XRD Analyses. The authors recognize and thank Dr. Khalil A. Khalil for editorial handling and Dr. Silvio R.F. Vlach and an anonymous reviewer for helpful comments on this version of the manuscript.

## Author contributions

**Conceptualization:** Megan E Holycross, Buz Barstow, Esteban Gazel.

**Data curation:** Joseph Brian Balta.

**Formal analysis:** Joseph Brian Balta.

**Funding acquisition:** Buz Barstow, Esteban Gazel.

**Investigation:** Joseph Brian Balta, Megan E Holycross.

**Methodology:** Joseph Brian Balta.

**Project administration:** Megan E Holycross, Esteban Gazel.

**Supervision:** Esteban Gazel.

**Writing – original draft:** Joseph Brian Balta.

**Writing – review & editing:** Megan E Holycross, Buz Barstow, Esteban Gazel.

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
