## [Decision Letter · Decision Letter 0]

24 Jul 2025

Dear Dr. Balta,

Thank you for submitting your manuscript to PLOS ONE. After careful consideration, we feel that it has merit but does not fully meet PLOS ONE’s publication criteria as it currently stands. Therefore, we invite you to submit a revised version of the manuscript that addresses the points raised during the review process.

We look forward to receiving your revised manuscript.

Kind regards,

Khalil Abdelrazek Khalil, Ph.D.

Academic Editor

PLOS ONE

Journal Requirements: 

 [This project was supported by ARPA-E award DE-AR0001341 to B.B, E.G., and M.E.H.]. 

4. Please expand the acronym “ARPA-E” (as indicated in your financial disclosure) so that it states the name of your funders in full.

Reviewers' comments:

Reviewer's Responses to Questions

**Comments to the Author**

1. Is the manuscript technically sound, and do the data support the conclusions?

Reviewer #1: Yes

Reviewer #2: Yes

2. Has the statistical analysis been performed appropriately and rigorously?

Reviewer #1: N/A

Reviewer #2: N/A

3. Have the authors made all data underlying the findings in their manuscript fully available?

Reviewer #1: Yes

Reviewer #2: Yes

4. Is the manuscript presented in an intelligible fashion and written in standard English?

Reviewer #1: Yes

Reviewer #2: Yes

Reviewer #1: Comments on the manuscript "Co-generation of NaREE(MoO4)2 and REEPO4 in multiple habits by solid-flux crystal growth".

The main goal of the paper is to refine the lab conditions for REE-inorganic phases synthesis. The focus is on the synthesis of REE phosphate (monazite) and of sodium-molybdenum-REE oxide (NaREE(MoO4)2) that has several industrial applications.

To synthesize the REE-oxide, the strategy was to separate REE from a REE-bearing phase (artificial Ce, Nd, Gd, Yb, La phosphates or natural monazite) using a mixture of an oxide (MoO3) and a carbonate (Na2CO3, LiCO3) as a flux, and subsequently varying i) the proportions of oxide:carbonate and phosphate:flux, ii) the temperature of loading, permanence and removal from heating in furnace, and iii) cooling in fast or slow conditions.

To achieve their goal, the authors based their research on previously published lab procedures and carried out four experiment types with successive degrees of improvement in procedures and results. Experiments 1 and 2 resulted in learning about the materials and mixtures used, the best way to heat and cool the samples, and the safe use of experimental apparatus (e.g., crucible, furnace). Experiment 3 had success in crystallizing up to 5 mm monazite. Experiment 4 was distinguished by its use of natural monazite as starting material and heat treatment at high pressures in a piston-cylinder apparatus and was successful in synthesizing large monazite surrounded by lathes of NaREE(MoO4)2.

The authors also provided updated recommendations for safer lab procedures and for synthesizing larger REEPO4 crystals of interest to the industrial sector.

The text is well-written and perfectly understandable. A few typing mistakes and punctuation errors were found (e.g., a full stop in line 139, a semicolon in line 225, and parentheses in line 589).

The four experiment types and new results are described in detail in the text. Samples were characterized by optical microscopy, Raman spectroscopy, XRD, electron microprobe, and laser ablation ICP-MS. SEM images of good quality illustrate the crystallized phases. The data provided by the study is of good quality and supports interpretations and conclusions.

The article is essential for improving the knowledge about the methods that can separate REE from minerals such as monazite, an abundant mineral containing REE. Thus, the manuscript fits the scope of the PLOS ONE journal. However, I would recommend minor modifications and clarifications before publication, as listed below.

Minor recommendations to the manuscript:

1) In line 146, “project” could be changed to “study, investigation”.

2) In line 152, “presentation” could be changed to “paper”.

3) The minerals mentioned in the captions of figures 4 and 5 could be labeled in the images.

Reviewer #2: I read your manuscript and think it brings some news to the subject and may be of interest to a significant audience, after some moderate/major optimization. I think your experimental and analytical data are mostly ok and your main conclusions seems to sound mostly well. Congrats! Please find in the below some suggestions/corrections and comments. Hope they help to improve the presented version. I am not native so apologize for any English Language fault.

Silvio RF Vlach

General: not a big deal, however normally the used abbreviations are explained in the first time they appear on the text; e.g. Electron Probe Micro Analyzer (EPMA). Please do not mix methods with results and/or discussion (e.g., Materials and Methods, Experiment type three: Successful phosphate synthesis)

Methods: the standards used for EPMA quantification need to be better referred; indicate also the method used for spectral treatment and matrix effect corrections as well as the expected detection limits/analytical errors. These latter need to be also informed in the section on LA-ICP-MS analysis.

When you mention flux : REEPO4 ratios please add in the table and at the first mention in the text if this ratio refers to weigh or molar ratios. How is the precision of the used analytical balance?

Results: the audience would like to see some representative compositions of the products (REEPO4 and NaREE(MoO4)2 crystals from the main experiments within the main text. Also how your LREE, P, Mo and Na analytical results from EDS compares with those from LA-ICP-MS? In general WDS (and so I think probably EDS) results for the major elements are better than those obtained with LA-ICP-MS. Along with such compositions, please add the relevant cationic proportions in relation to the appropriate Oxygen numbers to demonstrate the stoichiometry of your products. This will reinforce your XRD results and allow the audience to validate your analytical results on the main crystalline products.

You mention the use of the Raman spectroscopy but I did not see any results and the relevance of this method for them.

OK, the used flux avoids Pb contamination, but how about Mo contamination? If there is some, how is the advantage of the method in the case of synthesized REEPO4 standards?

Line 48: please, the so-called rare earth elements include, besides the lanthanides, Sc and Y (see IUPAC).

Line 53: think REE-bearing is better

Line 54-55: please, basnäsite is a LREE fluorocarbonate. I think basnäsite does not contain significant amounts of Th, U, and, so, Pb. PLease check!

Line 58: (a rock or mineral with economic, etc.)

Lines 95-96. I think you should also take a look in Donovan et al. (2002), J. Res. Natl. Inst. Stand. Technol., 107, and references therein for REE phosphate synthesis and Pb contamination trough Pb-bearing fluxes.

Line 111: are they all crystal habits or include crystal aggregate habits?

Line 139. …carbonate flux. (period) If this…

Line 166: please extend the table caption. Add P/F ratio = Phosphate/Flux (molar or weight) ratio in it. Specify better the experiments carried out with a furnace and a piston-cylinder. Add the simulation pressure in the latter case.

Line 343. Raman spectroscopy and X Ray Diffractometry sounds better. Analogous for the other presented methods.

Line 362. I think the readers would like to see some EDS compositional maps.

Line 364: please specify that the quantification was done with EDS (rather than WDS) analysis.

Line 635. Is Na3PO4 a reliable compound? Please check!

Line 642. Of note, the transition to a xenotime structure is significantly related to the cationic radii and should be expected in all REEPO4 compounds where REE has a relatively small radius, approaching the Y radius (as the most heavy lanthanides).

Line 654. I missed something here: “intermediate hexagonal structure”?

Line 683. I think the industrial potential and eventual drawbacks should be more elaborated in this paragraph.

Line 695: revise phrasing.

Lines 701-705. In some place along the text put some statements/comparisons on the relations between the obtained/known crystal habits and experimental conditions. Certainly there is a significant one.

Figures

Figure 1 caption: Ok, Nd-monazite crystallizes in the monoclinic system, however I think you cannot conclude this just on the basis of the habit of the shown crystal; a similar crystal should have also a triclinic symmetry. The last phrase is unnecessary.

Figure 2. I think XRD diffractogram/spectrum is better than XRD analyses. Sorry, I don’t remember: is it ok to indicate (1,1,10), with commas, as a Miller index? Please check.

Figure 3 caption: The samples L to R cannot be properly seen in (a); add letters below the respective containers. There is no tweezers in (b), just a conventional scale.

Figure 4. please specify: SE (secondary electron) images. I think you could say more based on them, concerning the observed crystalline habits. In (d) it appears you have a skeletal crystal (?). I am not sure if the NOTE is necessary.

Figure 5. Please identify (annotate) each phase over the BSE images.

Supplementary Materials

Please, optimize the organization and formats of your numeric tables to facilitate readers/potential users.

References

I did not check all in detail. However check the above suggested references.

**Do you want your identity to be public for this peer review?** For information about this choice, including consent withdrawal, please see our Privacy Policy

Reviewer #1: No

Reviewer #2: **Yes: ** Silvio Roberto Farias Vlach

---

## [Author Response · Author response to Decision Letter 1]

24 Sep 2025

Please see attachment for response to reviewers.

---

## [Decision Letter · Decision Letter 1]

8 Oct 2025

Co-generation of NaREE(MoO4)2 and REEPO4 in multiple habits by solid-flux crystal growth

PONE-D-25-29063R1

Dear Dr. Balta,

We’re pleased to inform you that your manuscript has been judged scientifically suitable for publication and will be formally accepted for publication once it meets all outstanding technical requirements.

Kind regards,

Khalil Abdelrazek Khalil, Ph.D.

Academic Editor

PLOS ONE

Additional Editor Comments (optional):

Reviewers' comments:

Reviewer's Responses to Questions

**Comments to the Author**

Reviewer #1: All comments have been addressed

Reviewer #2: All comments have been addressed

2. Is the manuscript technically sound, and do the data support the conclusions?

Reviewer #1: Yes

Reviewer #2: Yes

3. Has the statistical analysis been performed appropriately and rigorously?

Reviewer #1: Yes

Reviewer #2: N/A

4. Have the authors made all data underlying the findings in their manuscript fully available?

Reviewer #1: Yes

Reviewer #2: Yes

5. Is the manuscript presented in an intelligible fashion and written in standard English?

Reviewer #1: Yes

Reviewer #2: Yes

Reviewer #1: The authors have addressed the comments in the text and figures and the paper is recommended to publicaton.

Reviewer #2: Authors had addressed/responded my comments on the manuscript first draft in an appropriate fashion. So I recommend the publication of the final presented version as it is given its new and significant input.

**Do you want your identity to be public for this peer review?** For information about this choice, including consent withdrawal, please see our Privacy Policy

Reviewer #1: No

Reviewer #2: **Yes: ** SIlvio RF Vlach

---

## [Editor Report · Acceptance letter]

PONE-D-25-29063R1

PLOS ONE

Dear Dr. Balta,

I'm pleased to inform you that your manuscript has been deemed suitable for publication in PLOS ONE. Congratulations! Your manuscript is now being handed over to our production team.

Kind regards,

on behalf of

Dr. Khalil Abdelrazek Khalil

Academic Editor

PLOS ONE